# DGS: ROBUST AND DIVERSE WATERMARKS FOR DIFFUSION MODELS

## ABSTRACT

Recent advances in diffusion-based generative models, such as Stable Diffusion, have transformed image generation, making it possible to create high-quality and diverse content from textual prompts. However, these advancements also raise concerns about intellectual property theft and the authenticity of generated content. A promising solution to these issues is watermarking, which embeds hidden information into generated content to ensure traceability and protect intellectual property. In this paper, we propose Dynamic Gaussian Shading (DGS), a novel watermarking method specifically designed for diffusion models. DGS uses a dynamic, distance-aware re-localization approach for watermark embedding that adapts to the latent space of generative models, enhancing both the robustness of the watermark and the diversity of the generated images. We evaluate DGS in terms of its watermarking effectiveness, resistance to various attacks, and the diversity of generated images. Our experimental results show that DGS achieves high watermark accuracy, maintains robustness against attacks, and preserves image quality. Furthermore, we introduce a new metric, Encoded Feature Diversity (EFD), to measure the diversity of generated images across different watermarking methods. Compared to existing baseline methods, DGS strikes a significantly improved balance between watermark reliability and image generation diversity. The proposed method provides a promising approach to embedding watermarks in generative models, supporting the secure use of AI-generated content while maintaining the creative potential of these powerful tools.

## 1 INTRODUCTION

In recent years, deep learning-based LeCun et al. (2015) image generation models Song et al. (2020); Ho et al. (2020); Rombach et al. (2022); Nichol et al. (2021); Kingma (2013); Goodfellow et al. (2020); Ramesh et al. (2022); Razzhigaev et al. (2023), particularly those using diffusion processes, have shown remarkable progress in generating high-quality, diverse images from textual prompts. These models have become central to various applications, including artistic creation, drug design Huang et al. (2024); Guan et al. (2023), material design Xie et al. (2021); Li et al. (2022), and even data synthesis Zhu et al. (2024); Trabucco et al. (2023) for machine learning tasks. However, as with many powerful generative models, there are concerns regarding the intellectual property and authenticity of the generated content.

In recent years, deep learning-based image generation models, especially those using diffusion processes, have made significant strides in generating high-quality, diverse images from textual prompts. These advancements have been driven by models like Denoising Diffusion Probabilistic Models (DDPM) and its variants Song et al. (2020); Ho et al. (2020); Rombach et al. (2022); Nichol et al. (2021), along with others such as VAEs Kingma (2013), GANs Goodfellow et al. (2020), and hierarchical methods Ramesh et al. (2022); Razzhigaev et al. (2023). These models are now at the forefront of numerous applications, including artistic creation, drug design Huang et al. (2024); Guan et al. (2023), material design Xie et al. (2021); Li et al. (2022), and data synthesis Zhu et al. (2024); Trabucco et al. (2023) for machine learning tasks. However, the rise of these powerful generative models has also led to concerns about the intellectual property and authenticity of the generated content.

Recently, there has been growing concern over the misuse of content generated by these models. Typical forms of misuse include presenting AI-generated works as human-made for commercial purposes, or using generative models to plagiarize existing content. The ability to distinguish whether a piece of content has been generated by a model and to trace its origin has become a key issue in the field. To address these challenges, watermarking Cox et al. (2007); Wolfgang & Delp (1996) has emerged as a popular solution.

Watermarking, a classic technique used for protecting the intellectual property of works, has long been a mainstream approach Cox et al. (2007); Chang et al. (2005); Razzhigaev et al. (2023); Zhu (2018) to such issues. It involves embedding preset watermark information into a piece of content to resolve copyright disputes and ensure the work's authenticity. Compared to traditional watermarking techniques that directly hide watermarks within images, recent advances in diffusion-based watermarking methods Fernandez et al. (2023); Cui et al. (2023); Xiong et al. (2023); Zhao et al. (2023); Asnani et al. (2024); Jang et al. (2024) have shown superior robustness against various image manipulations, such as cropping, dropping, and noise addition. For instance, Wen et al. introduced the Tree-Ring method Wen et al. (2023), based on Fourier transforms, and Yang et al. proposed the Gaussian Shading method Yang et al. (2024), which uses distribution-preserving sampling. Both of these methods are plug-and-play and can be applied to any diffusion model without requiring modifications to the underlying model. They also maintain watermark effectiveness against a wide range of attacks while preserving the quality of generated images. However, our experiments show that these methods compromise the diversity of the generated images.

To address this limitation, we propose Dynamic Gaussian Shading (DGS), a novel watermarking method that improves upon the Gaussian Shading technique. In the Gaussian Shading method, the latent Gaussian space is divided into several non-overlapping subspaces, each corresponding to a specific watermark information. Sampling from a particular subspace embeds the watermark information associated with that subspace. We believe that this limitation of the fixed sampling space is the primary cause of the reduction in image diversity observed in this method. Based on this observation, we extend the original watermark by introducing randomly generated watermarks. These random watermarks shift the corresponding fixed subspace of the original watermark, allowing the subspaces to randomly change within the Gaussian space. This expansion of the sampling space increases the diversity of the generated images. Furthermore, we introduce a distance-weighted voting method in the watermark extraction phase, further improving the robustness of watermark detection.

We evaluate DGS through a series of experiments, comparing it with existing baseline methods in terms of watermark effectiveness, robustness against attacks, and image generation quality. Furthermore, we introduce a new metric, Encoded Feature Diversity (EFD), to assess the diversity of generated images across different watermarking methods. Our results show that DGS achieves superior watermark accuracy with minimal impact on image quality, while also maintaining high diversity in generated content. The proposed method offers a promising step forward in secure and diverse watermarking for generative models, opening up new possibilities for content protection in the age of AI-generated media. To the best of our knowledge, we are the first to investigate the impact of watermarking based on diffusion models on the diversity of generated images. We introduce a novel evaluation metric, Encoded Feature Diversity (EFD), which provides a fresh perspective on assessing diffusion model watermarking methods, beyond the traditional focus on image quality.

Our contributions can be summarized as follows:

- Introduction of Dynamic Gaussian Shading (DGS): We propose a novel watermarking method, DGS, which introduces a theoretically grounded distance-aware re-localization and distance-weighted voting mechanism to embed the watermark into a dynamically shifted latent subspace. These enhancement significantly increases the diversity of generated images while maintaining watermark robustness against various attacks.

- Encoded Feature Diversity (EFD): We introduce EFD, a new metric for evaluating the diversity of generated images under watermarking techniques. This metric provides a unique perspective, complementing traditional quality measures such as FID and CLIP scores.

- Comprehensive Evaluation of Watermarking Methods: Our work is the first to explore the effect of watermarking based on diffusion models on the diversity of generated images under a fixed prompt, filling a gap in the existing literature and providing insights into balancing watermark robustness and image diversity.

## 2 RELATED WORK

**Diffusion Models.** Recently, diffusion models have emerged as a powerful class of image generation techniques. Their foundation was laid by Sohl-Dickstein et al. Sohl-Dickstein et al. (2015), who introduced the concept of fitting data distributions through an inverse asymptotic noise process. This was further developed by Ho et al. Ho et al. (2020), who integrated the principles of score-based generative models Song & Ermon (2019; 2020) to propose Denoising Diffusion Probabilistic Models (DDPM). These models excel at generating high-quality images by sampling from an initial Gaussian distribution. More recently, Song et al. Song et al. (2020) introduced Denoising Diffusion Implicit Models (DDIM), which optimize the inverse process computation, allowing for image generation in fewer steps while maintaining quality.

**Image Watermarking.** Copyright protection has always been a hot topic of concern. For images, a popular method is to insert a watermark Cox et al. (2007); Wolfgang & Delp (1996) containing predefined information into the image. Traditional image watermarking methods, such as those Cox et al. (2007); Chang et al. (2005) based on Discrete Cosine Transform (DCT) and Discrete Wavelet Transform (DWT), have long been used for embedding information in images. However, the rise of generative models has driven the development of deep learning-based watermarking approaches, including adversarial methods Zhu (2018) and feature-based techniques Zhang et al. (2019). Although these methods show robustness to conventional attacks (e.g., compression and random cropping), they often struggle to embed watermarks in AI-generated content without degrading the quality or interpretability of the images.

**Watermarking for Diffusion Models.** As diffusion models became more widely adopted, researchers developed watermarking techniques uniquely suited to these models. Unlike traditional methods that embed watermarks directly into images, current watermarking techniques for diffusion models can be classified into two categories based on the watermark carrier. The first category involves embedding the watermark within the model itself. These methods Fernandez et al. (2023); Kim et al. (2024); Liu et al. (2023); Jiang et al. (2023) typically require fine-tuning the diffusion model or modifying its parameters. The second category Wen et al. (2023); Yang et al. (2024) adds the watermark to the initial latent variables, allowing for a plug-and-play approach that does not require any modifications to the underlying model, allowing compatibility with any diffusion model.

## 3 BACKGROUNDS

**DDIM Inversion.** Embedding watermark information directly into the initial latent variable $Z_T^s$ has become a popular approach in diffusion-based watermarking. In the watermark detection and extraction stages, it is essential to retrieve the initial latent variable $Z_T^{ls}$ corresponding to the target image. This process is the inverse of the typical denoising pathway in diffusion models; here, the target image is used as the input and the corresponding latent variable $Z_T^{ls}$ is the output.

Currently, the most widely used method for this inversion is the DDIM inversion Dhariwal & Nichol (2021). By estimating the additive noise at each step, DDIM Inversion can gradually reconstruct the initial latent $Z_T^s$, ensuring that the resulting $Z_T^{ls}$ closely approximates the original latent variable used during embedding.

**Gasussian Shading.** Gaussian Shading is a watermarking technique that embeds information directly into a model's latent variable. This is achieved by partitioning the Gaussian latent space into distinct, non-overlapping subspaces, with each subspace corresponding to a unique watermark. To enhance robustness, the watermark is repeated multiple times before being encrypted using the ChaCha20Bernstein et al. (2008) algorithm. This encryption step generates a uniformly distributed binary string of length $l$, ensuring that the probability distribution $p(y) = \frac{1}{2^l}$ is maintained. Finally, the initial latent variable $Z_T^s$ is sampled on the basis of this encoding, as specified in Eq. equation 1, $ppf$ is the quantile function of the Gaussian distribution and $F$ is the cumulative distribution function, so that any sample drawn from a given subspace inherently carries the associated watermark information.

$$Z_T^s = ppf(\frac{F(Z_T^s|y=i)+i}{2^l}). \tag{1}$$

This approach, known as Distribution-Preserving Sampling, ensures that the latent variable $Z_T^s$ is sampled exclusively within the subspace corresponding to a particular watermark, thereby preserving a predefined distribution as outlined in Eq. equation 2. This method secures the watermark's integrity by aligning $Z_T^s$ distribution with the watermark-specific subspace, enabling reliable and consistent watermark detection.

$$p(Z_T^s|y=i) = \begin{cases} 2^l * f(Z_T^s), & ppf(\frac{i}{2^l}) < Z_T^s < ppf(\frac{i+1}{2^l}), \\ 0, & \text{otherwise.} \end{cases} \tag{2}$$

## 4 THE PROPOSED METHOD

In this section, we begin by introducing the threat model considered in this work, outlining the potential risks and adversarial scenarios that our watermarking method aims to address. Following this, we provide a detailed explanation of the watermark embedding and extraction processes, highlighting how Dynamic Gaussian Shading (DGS) ensures robustness and diversity within these procedures.

### 4.1 THREAT MODEL

We begin by defining two roles: **the model owner**, who publishes an API for users to generate images using the model, and **the attacker**, a user who attempts to use generated images for prohibited purposes. The purpose of watermarking is twofold: to verify whether an image was generated by the model published by the model owner and to invisibly encode generation information (e.g., user ID).

In this setup, the model owner integrates the watermarking algorithm directly into the API, so that a watermark is embedded each time a user calls the API to generate an image. When an attacker generates an image, they may attempt to remove the watermark through attacks like image transformations and other methods before using the image in unauthorized ways.

If a dispute arises, the model owner can extract the watermark from the image for verification. The watermark typically has two components: (1) Fixed information that verifies the image as generated by the model owner's model, and (2) Dynamic information, such as unique user IDs, that identifies the specific user who generated the image.

### 4.2 WATERMARK EMBEDDING

We propose a method called **Distance-aware Re-localization**, which embeds watermarks by filling randomly generated noise into the latent variable. The overall pipeline is illustrated in Fig. 1(a).

We begin with the target watermark sequence $W_{\text{init}}$ and a set of randomly generated Gaussian noise samples $N$. The watermark $W_{\text{init}}$ is a binary string of length $l$, while $N$ contains $c \times h \times w$ Gaussian noise values, where $c$, $h$, and $w$ denote the channel and spatial dimensions of the latent variable.

Next, we randomly generate a binary sequence $W_{\text{rand}}$ of length $\frac{l}{f_l}$, where the hyperparameter $f_l$ controls the trade-off between watermark robustness and diversity. The initial watermark $W_{\text{init}}$ is then transformed by a shift function $M$, yielding a binary sequence $W_{\text{shift}}$ of equal length. Here, we implement $M$ as a broadcast XOR operation.

Following the Gaussian Shading paradigm, both $W_{\text{shift}}$ and $W_{\text{rand}}$ are expanded by repeating them $g_{\text{shift}}$ and $g_{\text{rand}}$ times, respectively. The resulting sequences are further transformed using the ChaCha20 algorithm to yield two encrypted binary sequences: $EW_{\text{shift}}$ of length $l \times g_{\text{shift}}$, and $EW_{\text{rand}}$ of length $\frac{l}{f_l} \times g_{\text{rand}}$. These lengths are chosen such that $l \times g_{\text{shift}} + \frac{l}{f_l} \times g_{\text{rand}} = c \times h \times w$. We then construct a binary mask sequence $Mask_{\text{rand}}$ from $W_{\text{rand}}$, with the same length as $EW_{\text{rand}}$. This mask is used to form a ternary sequence MEW, consisting of three symbols: **P**, **N**, and **R**. Specifically, if a bit in $Mask_{\text{rand}}$ equals 0, the corresponding entry in MEW is set to **R**; otherwise, its value is determined by $EW_{\text{shift}}$, where 1 maps to **P** and 0 maps to **N**.

Based on the number of **P**, **N**, and **R** entries in MEW, we split the Gaussian noise set $N$ into three subsets: the *positive set*, *negative set*, and *remaining set*. The partitioning is performed according to

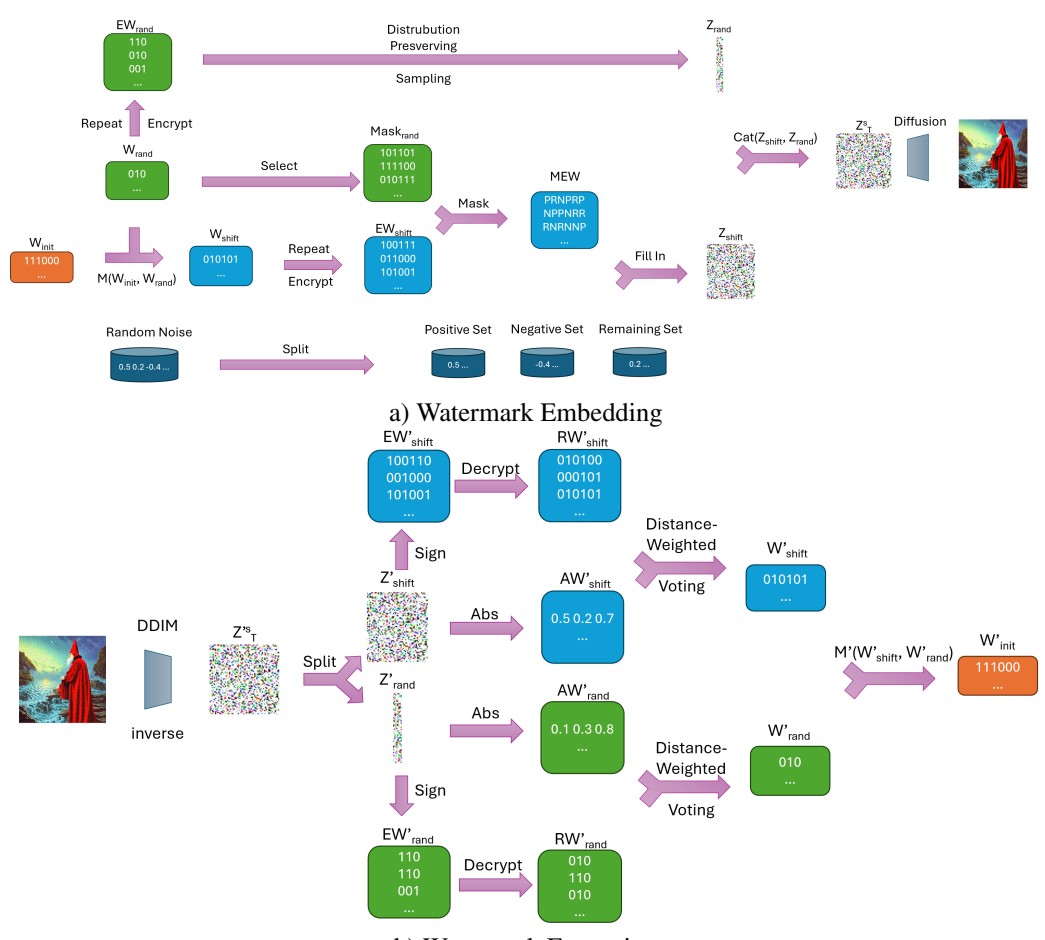

a) Watermark Embedding

b) Watermark Extraction

Figure 1: The pipeline of Dynamic Gaussian Shading. Distance-aware Re-localization splits Gaussian noise by absolute magnitude and fills it into the latent variable according to the watermark sequence and a random binary string $W_{\text{rand}}$, enabling diverse watermark embedding.

the absolute magnitude of noise values: large positive values are assigned to the positive set, large negative values to the negative set, and the smaller values to the remaining set.

Finally, we randomly sample noise from these three subsets and fill them into the positions specified by MEW, resulting in the noise sequence $Z_{\text{shift}}$. Meanwhile, the noise sequence $Z_{\text{rand}}$ is generated directly from $W_{\text{rand}}$ via Distribution-Preserving Sampling. Concatenating $Z_{\text{rand}}$ and $Z_{\text{shift}}$, and reshaping the result, yields the latent variable $Z_T^s$ that encodes the watermark.

Below we give an explanation of how Distance-aware Re-localization improves the robustness of watermark embedding. Our approach is inspired by an intriguing observation: the closer a sampled latent variable is to the boundary of its subspace, the more likely it is that, after an attack, the inverted latent variable will cross this boundary. To illustrate this, consider a simple example in a two-dimensional Gaussian space, divided into four equal subspaces by the coordinate axes (see Figure 2(a)). Here, both samples within the first subspace encode binary information "00". However, the green sample point is located very close to the boundary between the first and fourth subspaces. We hypothesize that different types of attack on the image will have a similar impact on both sample points, but, as shown in Figure 2(b) (the three images on the right), the green point is particularly prone to cross the boundary after inversion after various attacks.

Next, we give a theoretical proof. Consider $z_i$ as the value at the $i$-th position of the initial latent $Z$, and $z_i'$ as the corresponding value at the $i$-th position of the latent $Z'$ after a certain attack and subsequent DDIM reverse process. We compute the probability $P(\text{sign}(z_i) \neq \text{sign}(z_i'))$. Assuming

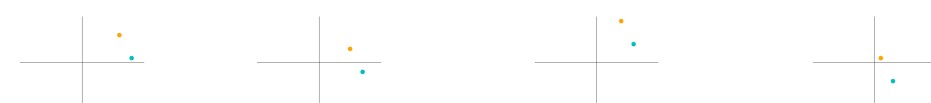

a) origin sampled latent                    b) reversed latent after attacked

Figure 2: The green and orange dots represent latent variables randomly sampled twice. The latent variables sampled close to the boundary line (the green ones) are more likely to change the subspace they belong to when the image is attacked.

that the specific attack modifies $z_i$ by a random variable $X$ following a probability distribution $X$, we express this as:

$$z'_i = z_i + X. \tag{3}$$

Let $f_X$ be the probability density function of $X$. Then, we have:

$$P(\text{sign}(z_i) \neq \text{sign}(z'_i)) = \begin{cases} \int_{-\infty}^{-z_i} f_X(x)\,dx, & \text{if } z_i > 0 \\ \int_{-z_i}^{\infty} f_X(x)\,dx, & \text{if } z_i < 0 \end{cases} \tag{4}$$

Since $f_X$ is non-negative, as the absolute value of $z_i$ increases, meaning the distance of $z_i$ to the boundary of a certain dimension in space increases, the probability $P(\text{sign}(z_i) \neq \text{sign}(z'_i))$ decreases. This implies that $z_i$ exhibits greater robustness in preserving its sign against arbitrary attacks.

In the following, we explain how shifting $W_{\text{init}}$ using the randomly generated $W_{\text{rand}}$ and the function $M$ influences the diversity of the generated latent variable $Z_s^T$. Starting with a fixed $W_{\text{init}}$, as described in the threat model, each user is assigned a unique watermark for identification. This watermark remains constant whenever the user generates an image via the API. Given the uniform distribution of the encrypted watermark, for a fixed $W_{\text{init}}$, we can consider the encrypted watermark $EW_{\text{shift}}$ as a specific integer $k$ such that $k \in [0, 2^l)$, corresponding to the $k$-th Gaussian subspace out of the $2^l$ subspaces.

Now, by randomly generating a uniformly distributed binary string $W_{\text{rand}}$, which can be regarded as an integer in the range $\left[0, 2^{\frac{l}{f_l}}\right)$, and applying the shift function $M$ to $W_{\text{init}}$, the resulting $W_{\text{shift}}$ will follow a uniform distribution, as shown in Eq. equation 5.

$$p(y = i) = \begin{cases} \frac{1}{2^{\frac{l}{f_l}}}, & i \in M(k, u), \\ 0, & \text{otherwise.} \end{cases} \tag{5}$$

In this way, the subspace corresponding to $W_{\text{shift}}$ can dynamically vary with different values of $W_{\text{rand}}$. For a fixed $W_{\text{init}}$, this results in an upgrade from a single fixed subspace to a dynamic range of $2^{\frac{l}{f_l}}$ subspaces. In other words, the range from which the latent variable $Z_T^s$ can be sampled expands by a factor of $2^{\frac{l}{f_l}}$. At the same time, different $W_{\text{rand}}$ values correspond to different positions for assigning free elements from the remaining set. These elements do not contain watermark information, which further increases the size of the subspace corresponding to $W_{\text{shift}}$. In the next section, our experimental results will demonstrate the significant improvement this brings to the diversity of generated images.

### 4.3 WATERMARK EXTRACTION

The watermark embedding pipeline is illustrated in Figure 1(b). The watermark extraction process mirrors the embedding procedure. First, the image undergoes a DDIM inversion to retrieve the latent variable $Z_T^{\prime s}$. Following the positions used in the watermark embedding's concatenation process, we split the latent variable into two parts $Z'_{shift}$ and $Z'_{rand}$ to extract both the randomly shifted watermark and the randomly generated watermark. The extraction process for each part is identical, based on identifying the subspace corresponding to each segment of the split variable. The decrypted $RW'_{shift}$ and $RW'_{rand}$ then correspond to the repeated structure $RW_{shift}$ and $RW_{rand}$ used in the

embedding process. Conventionally, a voting-like method reverses this repetition to recover the original watermark $W'_{shift}$ and $W'_{rand}$. To enhance extraction accuracy, we introduce a distance-weighted voting method, which assigns weights based on distance, further refining the accuracy of the extracted watermark.

Based on previous theory, similarly, we can compute $P(\text{sign}(z_i) \neq \text{sign}(z'_i))$ using $z'_i$:

$$P(\text{sign}(z_i) \neq \text{sign}(z'_i)) = \begin{cases} \int_{z'_i}^{\infty} f_X(x)\, dx, & \text{if } z'_i > 0 \\ \int_{-\infty}^{z'_i} f_X(x)\, dx, & \text{if } z'_i < 0 \end{cases} \tag{6}$$

Since a larger absolute value of $z'_i$ increases the probability of maintaining the sign of $z_i$, we enhance the robustness against various attacks by introducing the distance to the boundary as a weighted sum when computing the vote. This process is illustrated in Eq. equation 7, using $W'_{shift}$ as an example. For a two-dimensional binary matrix $RW'_{shift}$ with dimensions $l \times g$, where l is the length of the initial watermark $W'_{shift}$, and $g$ represents the number of groups in the repeat operation, we perform a reverse reduction on $RW'_{shift[i,j]}$ across groups to compute the corresponding $W'_{shift[i]}$. Notably, $Z'_{shift}$ and $RW'_{shift}$ are mapped one-to-one, so we assign weights to their absolute values based on the distance from $Z'_{shift}$ to each boundary line. Notably, $RW'_{shift}$ contains free elements from the remaining set that do not include watermark information. To mitigate their influence, we first reconstruct $W'_{rand}$ and then set the corresponding positions' weights to zero based on $W'_{rand}$.

$$W'_{\text{shift}[i]} = \begin{cases} 1, & \sum_{j=1}^{g} |Z'_{\text{shift}[i,j]}| * RW'_{\text{shift}[i,j]} > \sum_{j=1}^{g} \left| \dfrac{Z'_{\text{shift}[i,j]}}{2} \right|, \\ 0, & \text{otherwise.} \end{cases} \tag{7}$$

## 5 EXPERIMENTS

In this section, we present analyses of DGS's watermark effectiveness, robustness against various attacks, diversity in generated images, and ablation studies.

### 5.1 WATERMARK EFFECTIVENESS AND IMAGE GENERATION QUALITY

In this subsection, we compare the propsoed DGS with the baselines. we selected DwtDct Cox et al. (2007), DwtDctSvd Cox et al. (2007), RivaGAN Zhang et al. (2019), Stable Signature Fernandez et al. (2023), Tree-Ring Wen et al. (2023), and Gaussian Shading Yang et al. (2024) for comparison. Specifically the average bit accuracy and TPR after applying the nine attacks mentioned above, as well as the quality of the generated images in terms of FID Heusel et al. (2017) and CLIP scores Radford et al. (2021).

The average bit accuracy corresponds to the localization requirement in Section 4.1's threat model, where dynamic watermarks are extracted from images to locate relevant information. We randomly generate 1,000 sets of watermarks, embed each watermark into images using each method, apply attacks to these images, and then extract the watermark from each attacked image. By comparing the extracted watermarks bit-by-bit with the original watermark, we calculate the overall average accuracy. Since the Tree-Ring method cannot extract watermarks, we also compute TPR as a baseline comparison. TPR corresponds to the first requirement in the throat model, which is to detect whether an image contains a fixed preset information watermark.

For image quality, we follow previous work using FID and CLIP score metrics. FID is calculated using 5,000 images from the COCO dataset Lin et al. (2014) and their corresponding prompts as ground truth, generating images with the same prompts to compute FID similarity. CLIP score measures the cosine similarity between the generated image and its prompt in the CLIP model's Cherti et al. (2023) encoded space.

The experimental results are shown in Table 1. Since the experimental setup is identical in this section, we directly copied the baseline results from Gaussian Shading. From the results, we can see that, in terms of watermark effectiveness, DGS achieved the highest average bit accuracy, improving

| Methods | TPR (Clean) | TPR (Adversarial) | Bit Acc. (Clean) | Bit Acc. (Adversarial) | FID | CLIP-Score |
|---|---|---|---|---|---|---|
| Stable Diffusion | - | - | - | - | 25.23±.18 | 0.3629±.0006 |
| DwtDct Cox et al. (2007) | 0.825/0.881/0.866 | 0.172/0.178/0.173 | 0.8030/0.8059/0.8023 | 0.5696/0.5671/0.5622 | 24.97±.19 | 0.3617±.0007 |
| DwtDctSvd Cox et al. (2007) | 1.000/1.000/1.000 | 0.597/0.594/0.599 | 0.9997/0.9987/0.9987 | 0.6920/0.6868/0.6905 | 24.45±.22 | 0.3609±.0009 |
| RivaGAN Zhang et al. (2019) | 0.920/0.945/0.963 | 0.697/0.697/0.706 | 0.9762/0.9877/0.9921 | 0.8986/0.9124/0.9019 | 24.24±.16 | 0.3611±.0009 |
| Tree-Ring Wen et al. (2023) | 1.000/1.000/1.000 | 0.894/0.898/0.906 | - | - | 25.43±.13 | 0.3632±.0006 |
| Stable Signature Fernandez et al. (2023) | 1.000/1.000/1.000 | 0.502/0.505/0.496 | 0.9987/0.9978/0.9979 | 0.7520/0.7472/0.7500 | 25.45±.18 | 0.3622±.0027 |
| Gaussian Shading Yang et al. (2024) | 1.000/1.000/1.000 | **0.997/0.998/0.996** | 0.9999/0.9999/0.9999 | 0.9753/0.9749/0.9724 | 25.20±.22 | 0.3631±.0005 |
| DGS (Ours) | **1.000/1.000/1.000** | 0.994/0.993/0.993 | **1.0000/1.0000/1.0000** | **0.9852/0.9838/0.9833** | 25.24±.12 | **0.3655±.0004** |

Table 1: Our proposed DGS is capable of embedding large-capacity watermarks without compromising image generation quality, and it maintains robustness against various attacks.

by 1.1% on average over Gaussian Shading, which was ranked second. The TPR of DGS is slightly lower than that of Gaussian Shading, averaging 0.3%, but remains highly reliable with a TPR greater than 99% in verification scenarios.

In terms of image generation quality, DGS performs comparably to other diffusion-based watermarking methods. In the FID test, DGS only scores 0.01 higher than standard Stable Diffusion, outperforming Tree-ring and Stable Signature, but slightly behind Gaussian Shading. However, in the CLIP score test, DGS achieves the highest score, surpassing standard Stable Diffusion by 0.26%, and outperforming Tree-ring, Stable Signature, and Gaussian Shading.

## 5.2 DIVERSITY OF GENERATED IMAGES

In the previous subsection, we observed that diffusion-based watermarking methods achieve high watermark effectiveness while barely affecting image generation quality, compared to directly embedding watermarks in images. It may appear that these methods allow watermark embedding with no trade-off. In this subsection, we reveal the trade-off through experiments, showing that these methods incur a loss in diversity among generated images.

Inspired by the CLIP score Radford et al. (2021) calculation, we propose the Encoded Feature Diversity (EFD) metric to evaluate the diversity of images generated by each method. First, we randomly generate a watermark, fixing it to generate n images for each method, under a **fixed prompt** and n random seeds. Then, we obtain features for each image via the CLIP model's Cherti et al. (2023) encoder. Unlike in CLIP score calculation, we compute the cosine distance between each pair of features. As shown in Eq. equation 8, the mean of all pairwise cosine distances serves as a measure of image diversity, where $X$ represents the generated image and $\varepsilon$ denotes the CLIP encoder. When the generated images are more diverse, the features are more dispersed, resulting in a higher mean cosine distance, i.e., a higher EFD.

$$EFD = \frac{1}{n^2} \sum_{i=1}^{n} \sum_{j=1}^{n} cosine - distance(\varepsilon(X_i), \varepsilon(X_j)). \tag{8}$$

For all the methods to be compared, we use the diffusion model version 2.1 and generate 1,000 images with exactly the same prompt and watermark to simulate the scenario of a single user repeatedly generating images. The experimental results are shown in Table 2. The normal Stable Diffusion has an EFD of 0.213, while methods such as DwtDct, DwtDctSvd, and RivaGAN show only a slight decrease in EFD. This is because they directly embed the watermark in the image without affecting the diffusion model generation process.

Among the diffusion-based watermarking methods, DGS achieves the highest EFD. It reduces the EFD by 0.075% compared to Stable Diffusion, while improving by 3.11% compared to Gaussian Shading. The results demonstrate that DGS provides a significant improvement in image diversity while maintaining watermark effectiveness and image generation quality comparable to Gaussian Shading.

Based on the above experimental data, it is evident that the current mainstream watermarking methods have minimal impact on image generation quality. The main trade-off lies between watermark effectiveness and image diversity. Methods like DwtDct, DwtDctSvd, and RivaGAN, which directly embed the watermark into the image, maintain high image diversity but lose robustness when subjected to attacks. On the other hand, diffusion model-based watermarking methods achieve robustness against various attacks at the cost of sacrificing image diversity. Besides providing a clearer

| Methods | EFD |
|---------|-----|
| Stable Diffusion | 0.21337326 |
| DwtDct Cox et al. (2007) | 0.21342951 |
| DwtDctSvd Cox et al. (2007) | 0.20842214 |
| RivaGAN Zhang et al. (2019) | 0.20892236 |
| Tree-Ring Wen et al. (2023) | 0.19838478 |
| Gaussian Shading Yang et al. (2024) | 0.18146528 |
| DGS (Ours) | 0.21261603 |

Table 2: Our proposed DGS achieves the highest EFD among the diffusion-based watermarking methods.

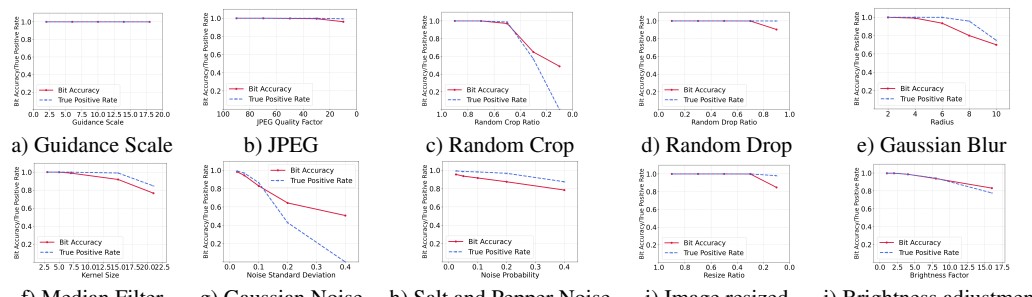

a) Guidance Scale    b) JPEG    c) Random Crop    d) Random Drop    e) Gaussian Blur

f) Median Filter    g) Gaussian Noise    h) Salt and Pepper Noise    i) Image resized    j) Brightness adjustment

Figure 3: Ablation studies results.

numerical trade-off, EFD is also more meaningful in real-world applications compared to FID and CLIP score. In practical generative model applications, an individual user cannot perceive the overall quality of images generated for all users but can only assess the images produced from their own prompts. Since EFD is computed based on a fixed user ID, improvements in EFD directly enhance the user experience.

## 5.3 ABLATION STUDIES

In this section, we conduct ablation experiments to evaluate the performance of DGS under various key parameters, including guidance scales and attack scales.

**Guidance Scales.** Guidance scales control the balance between the importance of the text condition. Following the setup used in Tree-Ring Wen et al. (2023), we varied the guidance scales from 2 to 18. The results are shown in Figure 3 (a). DGS consistently maintains high watermark effectiveness, even when the guidance scale is set to 18, where the use of an empty string as a prompt during DDIM inversion introduces minor errors that do not affect DGS robustness.

**Attack Scales.** The scale of an attack is an important hyperparameter, as larger attack scales pose a greater challenge to watermark robustness. We followed the attack settings from Gaussian Shading Yang et al. (2024) and tested each attack with increasing scales. The results are shown in Figure 3 (b)-(j). Except for more destructive attacks, such as random crop and Gaussian noise, DGS successfully defends against large-scale attacks from other methods.

## 6 CONCLUSION

In this paper, we propose Dynamic Gaussian Shading (DGS), a diffusion model watermarking method that strikes a balance between watermark effectiveness and image generation diversity. Our experimental results demonstrate that DGS can maintain watermark robustness against various attacks while preserving both image generation quality and diversity. In particular, DGS maintains over 98.3% accuracy in watermark extraction even under various attack scenarios, while achieving competitive performance in terms of image quality (FID and CLIP scores). Compared to baseline methods, DGS shows significant improvements in image diversity without sacrificing watermark reliability. Additionally, we introduce Encoded Feature Diversity (EFD) , the first metric to evaluate the impact of watermarking on the diversity of images generated by diffusion models, which enhances the assessment of diffusion model watermarking methods.

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

# A APPENDIX

This document provides additional experimental results and visualizations. First, we present the TPR and bit accuracy results of DGS under individual attacks. Next, we offer additional ablation experiments, comparing the performance of DGS and Gaussian Shading using Edict for inference and inversion. Finally, we include various visualizations to intuitively demonstrate the advantages of DGS.

## A.1 EXPERIMENTAL SETUPS

Following the setup used in Gaussian Shading Yang et al. (2024), all experiments were conducted on a single NVIDIA RTX 3090 GPU. The diffusion model used was the text-to-image model Rombach et al. (2022) provided by Hugging Face, specifically versions V1.4, V2.0, and V2.1. The image generation used 50 denoising steps with DPM Solver Lu et al. (2022) as the denoising method, using prompts from Stable-Diffusion-Prompt. For the reverse process, we also set 50 steps using the DDIM inversion Song et al. (2020) with an empty string as a prompt. The watermark consists of a binary string with a capacity of 256 bits, and we calculate both the average accuracy per bit and the true positive rate (TPR) at a fixed false positive rate (FPR) of $10^{-6}$.

The generated images are sized $512 \times 512$. Following previous works Yang et al. (2024); Wen et al. (2023), we calculate the FID Heusel et al. (2017) and CLIP Radford et al. (2021) scores to assess generation quality.

For the watermark robustness experiments, the attack methods and parameter settings are as follows:

1. JPEG Compression with Quality Factor (QF) set to 25.

2. Random crop applied to 60% of the image area.

3. Random drop applied to 80% of the image area.

4. Gaussian blur with a radius ($r$) of 4 pixels.

5. Median filter with a kernel size ($k$) of $7 \times 7$ pixels.

6. Gaussian noise with mean ($\mu$) of 0 and standard deviation ($\sigma$) of 0.05.

7. Salt and Pepper Noise with a probability ($p$) of 0.05 for each pixel to be altered.

8. Image resized to 25% of its original size, then restored to the original size.

9. Brightness adjustment with a factor of 6.

| Noise | Methods | | | | | | |
|---|---|---|---|---|---|---|---|
| | DwtDct Cox et al. (2007) | DwtDctSvd Cox et al. (2007) | RivaGAN Zhang et al. (2019) | Tree-Ring Wen et al. (2023) | Stable Signature Fernandez et al. (2023) | Gaussian Shading Yang et al. (2024) | Ours |
| None | 0.825/0.881/0.866 | 1.000/1.000/1.000 | 0.920/0.945/0.963 | 1.000/1.000/1.000 | 1.000/1.000/1.000 | 1.000/1.000/1.000 | 1.000/1.000/1.000 |
| JPEG | 0/0/0 | 0.013/0.019/0.015 | 0.156/0.085/0.214 | 0.997/1.000/0.994 | 0.210/0.217/0.198 | 0.999/1.000/0.997 | 1.000/1.000/0.999 |
| RandCr | 0.982/0.967/0.952 | 1.000/0.998/0.999 | 0.868/0.878/0.891 | 0.997/1.000/0.998 | 1.000/0.998/0.993 | 1.000/1.000/1.000 | 1.000/1.000/1.000 |
| RandDr | 0/0/0 | 0/0/0 | 0.887/0.885/0.862 | 1.000/1.000/0.998 | 0.971/0.980/0.972 | 1.000/1.000/1.000 | 1.000/1.000/0.999 |
| GauBlur | 0/0/0.001 | 0.430/0.419/0.432 | 0.328/0.331/0.316 | 1.000/1.000/0.997 | 0/0/0 | 1.000/1.000/1.000 | 1.000/1.000/1.000 |
| MedFilter | 0/0/0.001 | 0.996/0.999/1.000 | 0.863/0.832/0.873 | 1.000/1.000/1.000 | 0.001/0/0 | 1.000/1.000/1.000 | 1.000/1.000/1.000 |
| GauNoise | 0.354/0.353/0.364 | 0.842/0.862/0.884 | 0.441/0.457/0.535 | 0/0.006/0.077 | 0.424/0.406/0.404 | 0.996/0.995/0.995 | 0.968/0.978/0.976 |
| S&PNoise | 0.089/0.160/0.102 | 0/0.005/0.008 | 0.477/0.411/0.431 | 0.972/0.986/0.994 | 0.072/0.078/0.052 | 1.000/0.998/0.997 | 0.985/0.991/0.991 |
| Resize | 0/0.005/0.008 | 0.985/0.977/0.983 | 0.850/0.886/0.887 | 1.000/1.000/1.000 | 0/0/0 | 1.000/1.000/1.000 | 1.000/1.000/1.000 |
| Brightness | 0.126/0.114/0.124 | 0.110/0.072/0.074 | 0.480/0.404/0.386 | 0.084/0.089/0.092 | 0.843/0.862/0.849 | 0.974/0.991/0.979 | 0.989/0.973/0.975 |
| Average of Adversarial | 0.172/0.178/0.173 | 0.597/0.594/0.599 | 0.697/0.697/0.706 | 0.894/0.898/0.906 | 0.502/0.505/0.496 | 0.997/0.998/0.996 | 0.994/0.993/0.993 |

Table 3: results of TPR under each attack. DGS demonstrates slightly lower TPR compared to Gaussian Shading.

| Noise | Methods | | | | | |
|---|---|---|---|---|---|---|
| | DwtDct Cox et al. (2007) | DwtDctSvd Cox et al. (2007) | RivaGAN Zhang et al. (2019) | Stable Signature Fernandez et al. (2023) | Gaussian Shading Yang et al. (2024) | Ours |
| None | 0.8030/0.8059/0.8023 | 0.9997/0.9987/0.9987 | 0.9762/0.9877/0.9921 | 0.9987/0.9978/0.9949 | 0.9999/0.9999/0.9999 | 1.0000/1.0000/1.0000 |
| JPEG | 0.5018/0.5047/0.5046 | 0.5197/0.5216/0.5241 | 0.7943/0.7835/0.8181 | 0.7901/0.7839/0.7893 | 0.9918/0.9905/0.9872 | 0.9941/0.9941/0.9940 |
| RandCr | 0.7849/0.7691/0.7673 | 0.8309/0.7942/0.8151 | 0.9761/0.9723/0.9735 | 0.9933/0.9903/0.9883 | 0.9803/0.9747/0.9669 | 0.9987/0.9980/0.9977 |
| RandDr | 0.5540/0.5431/0.5275 | 0.5814/0.5954/0.6035 | 0.9678/0.9720/0.9683 | 0.9768/0.9747/0.9736 | 0.9676/0.9687/0.9649 | 0.9983/0.9978/0.9965 |
| GauBlur | 0.5000/0.5027/0.5039 | 0.6579/0.6466/0.6459 | 0.8323/0.8538/0.8368 | 0.4137/0.4110/0.4112 | 0.9874/0.9846/0.9858 | 0.9948/0.9943/0.9940 |
| MedFilter | 0.5171/0.5243/0.5199 | 0.9208/0.9287/0.9208 | 0.9617/0.9585/0.9696 | 0.6374/0.6399/0.6587 | 0.9987/0.9970/0.9990 | 0.9993/0.9994/0.9998 |
| GauNoise | 0.6502/0.6294/0.6203 | 0.7960/0.7950/0.8159 | 0.8404/0.9648/0.8776 | 0.7831/0.7766/0.7768 | 0.9636/0.9556/0.9592 | 0.9460/0.9460/0.9528 |
| S&PNoise | 0.5784/0.6021/0.5845 | 0.5120/0.5267/0.5250 | 0.8881/0.8838/0.8634 | 0.7192/0.7170/0.7144 | 0.9406/0.9433/0.9385 | 0.9491/0.9530/0.9491 |
| Resize | 0.5067/0.5184/0.5135 | 0.8743/0.8498/0.8630 | 0.9602/0.9731/0.9733 | 0.5278/0.5051/0.5177 | 0.9970/0.9975/0.9976 | 0.9981/0.9993/0.9993 |
| Brightness | 0.5336/0.5097/0.5175 | 0.5346/0.5234/0.5016 | 0.8666/0.8496/0.8369 | 0.9276/0.9267/0.9204 | 0.9508/0.9623/0.9527 | 0.9815/0.9718/0.9667 |
| Average of Adversarial | 0.5696/0.5671/0.5622 | 0.6920/0.6868/0.6905 | 0.8986/0.9124/0.9019 | 0.7520/0.7472/0.7500 | 0.9753/0.9749/0.9724 | 0.9852/0.9838/0.9833 |

Table 4: results of Bit accuracy under each attack. DGS achieves the best Bit accuracy.

## A.2 ADDITIONAL EXPERIMENTS

Tables 3 and 4 present the TPR and bit accuracy of DGS and other baselines under individual attack scenarios. As the experimental settings are identical, the baseline results are reproduced from Gaussian Shading. From the results, it can be observed that the performance under individual attacks is consistent with the overall averages. DGS demonstrates slightly lower TPR compared to Gaussian Shading but achieves the best bit accuracy overall. Notably, while Gaussian Shading performs worse than Tree Ring in Random Cropping and Random Dropping scenarios, DGS completely overcomes this limitation, achieving over 99

Since the watermark information is embedded into the initial latent variables, the accuracy of reconstructing the image back into latent variables is critical. In prior works, DDIM has been consistently used as the inverse algorithm without ablation studies. Here, we select the eDICT method as both the inference and inverse algorithm for ablation experiments and additionally evaluate the performance of Gaussian Shading as a comparison. Tables 5 and 6 present the experimental results for TPR and bit accuracy under these settings. The experimental results reveal a consistent trend: DGS exhibits slightly lower TPR than Gaussian Shading but achieves higher bit accuracy. A comparison of the results in Tables 1 and 2 demonstrates that improvements in reverse accuracy lead to a more significant enhancement in the watermarking effectiveness of DGS compared to Gaussian Shading.

## A.3 VISUAL RESULTS

To visually demonstrate the diversity enhancement of DGS compared to Gaussian Shading, we designed a series of visualization experiments. Figure 4 shows the visualization of 1,000 latent variables generated under the fixed watermark setting for both DGS and Gaussian Shading. Using PCA, the high-dimensional latent variables were reduced to two dimensions for display. As shown, the latent variable sampling subspace of DGS is significantly broader than that of Gaussian Shading.

Figure 5 visualizes the features of the corresponding images in the CLIP-encoded space, with PCA applied again to reduce feature dimensions to two for illustration. The results indicate that DGS produces more dispersed feature distributions compared to Gaussian Shading, reflecting a higher EFD score and showcasing more diverse image generation.

Figure 6 and 7 illustrate the visual comparison of image generation diversity between DGS and Gaussian Shading using prompts from the MS-COCO 2017 dataset. To provide a clearer depiction of the diversity, we applied a fixed watermark and restricted the sampling of latent variables to a narrow, randomly generated plane. This ensures that all latent variables are sampled from the intersection of this plane and the subspace corresponding to the watermark.

| Noise | Methods | |
|---|---|---|
| | Gaussian ShadingYang et al. (2024) | Ours |
| None | 1.000 | 1.000 |
| JPEG | 1.000 | 1.000 |
| RandCr | 1.000 | 1.000 |
| RandDr | 1.000 | 1.000 |
| GauBlur | 1.000 | 1.000 |
| MedFilter | 1.000 | 1.000 |
| GauNoise | 0.996 | 0.984 |
| S&PNoise | 0.999 | 0.996 |
| Resize | 1.000 | 1.000 |
| Brightness | 0.975 | 0.969 |
| Average of Adversarial | 0.9966 | 0.9943 |

Table 5: results of TPR under each attack using Edict for Inference and Inversion. Improved reverse methods can significantly enhance the watermarking effectiveness of DGS.

| Noise | Methods | |
|---|---|---|
| | Gaussian ShadingYang et al. (2024) | Ours |
| None | 1.0000 | 1.0000 |
| JPEG | 0.9538 | 0.9628 |
| RandCr | 0.9767 | 0.9969 |
| RandDr | 0.9715 | 0.9970 |
| GauBlur | 0.9862 | 0.9927 |
| MedFilter | 0.9990 | 0.9994 |
| GauNoise | 0.9659 | 0.9586 |
| S&PNoise | 0.9301 | 0.9372 |
| Resize | 0.9988 | 0.9994 |
| Brightness | 0.9538 | 0.9628 |
| Average of Adversarial | 0.9745 | 0.9819 |

Table 6: results of Bit accuracy under each attack using Edict for Inference and Inversion

As shown in the figure, under identical constraints and random seeds, Gaussian Shading produces images that are very similar, with only minor differences. In contrast, DGS generates significantly more diverse images, highlighting its ability to enhance diversity.

### A.4 LLM USE DECLARATION

Large Language Models (ChatGPT) were used exclusively to improve the clarity and fluency of English writing. They were not involved in research ideation, experimental design, data analysis, or interpretation. The authors take full responsibility for all content.

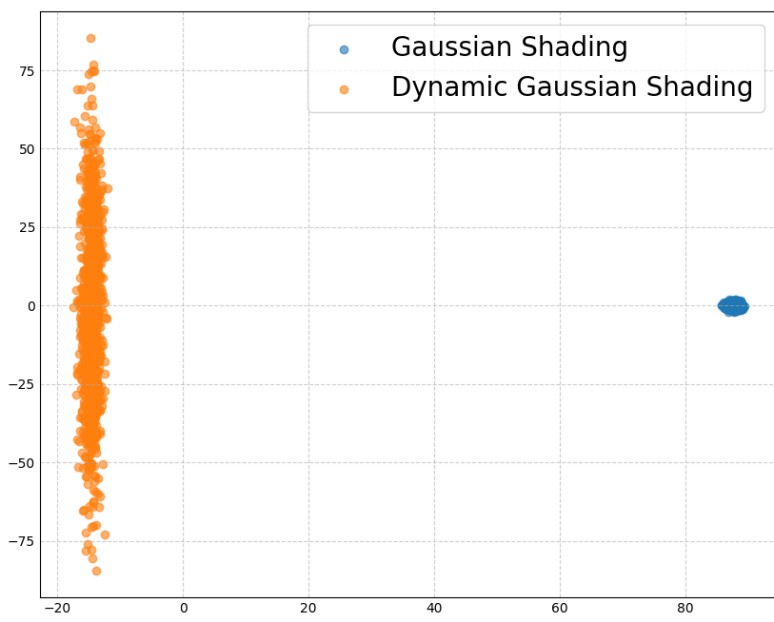

Figure 4: The visualization of latent variable distributions demonstrates that DGS exhibits greater diversity compared to Gaussian Shading.

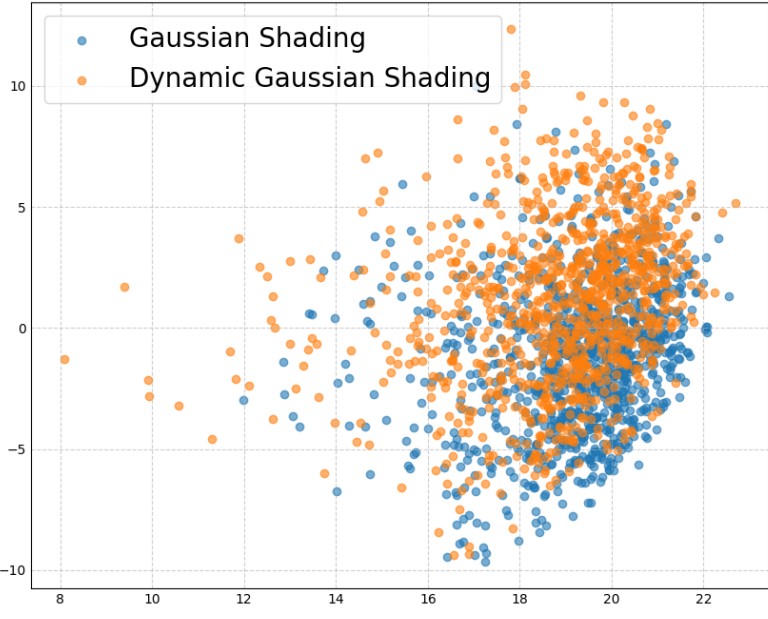

Figure 5: The visualization of generated image feature distributions shows that DGS achieves greater diversity compared to Gaussian Shading.

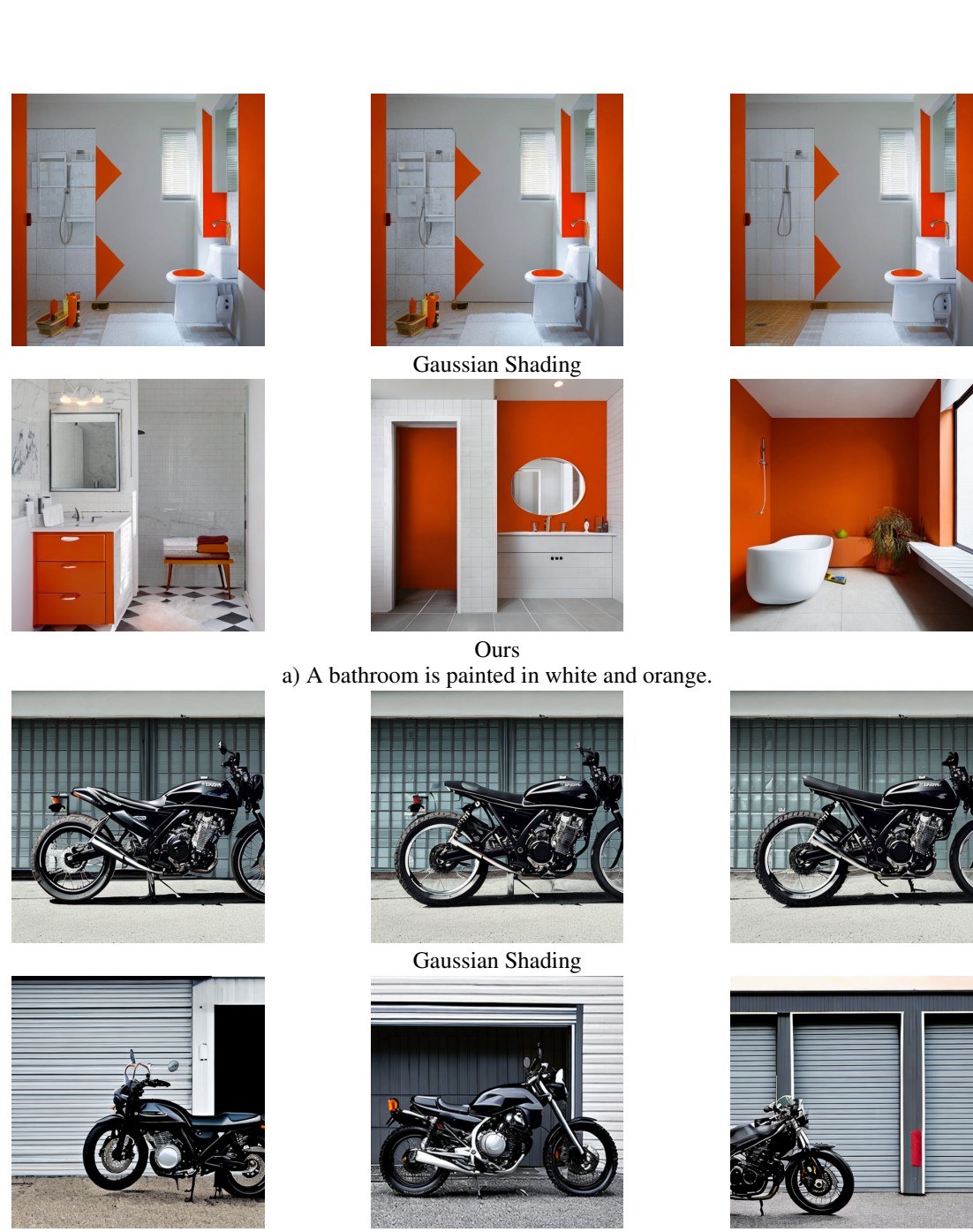

Gaussian Shading

Ours

a) A bathroom is painted in white and orange.

Gaussian Shading

Ours

b) A black Honda motorcycle parked in front of a garage

Figure 6: The visualization of images generated by DGS and Gaussian Shading under prompts from the MS-COCO2017 dataset demonstrates the diversity in outputs between the two methods. (Part 1)

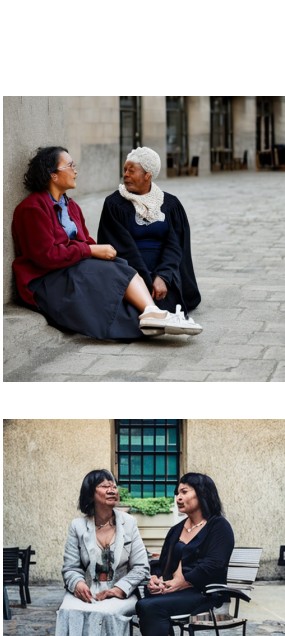
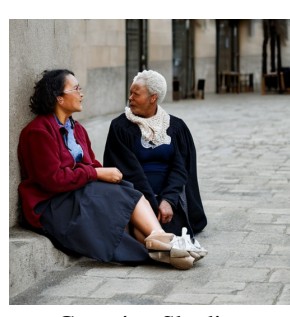
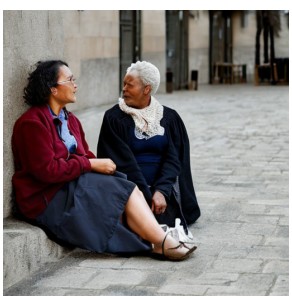

Gaussian Shading

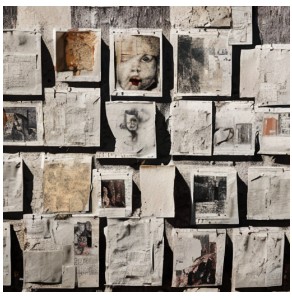
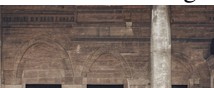
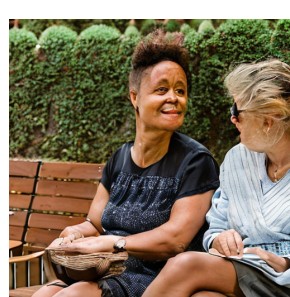

Ours

c) A couple of women sitting outside in a court yard.

Gaussian Shading

Ours

d) A plaster external wall with multiple old paper images attached.

Figure 7: The visualization of images generated by DGS and Gaussian Shading under prompts from the MS-COCO2017 dataset demonstrates the diversity in outputs between the two methods. (Part 2)

