# OpenReview forum: "DGS: Robust and Diverse Watermarks for Diffusion Models"
_ICLR.cc/2026/Conference — ICLR 2026 Conference Withdrawn Submission_

### Official Review · Reviewer_MADc · 2025-10-26

**Soundness:** 2
**Presentation:** 3
**Contribution:** 2
**Rating:** 2
**Confidence:** 4

**Summary:**

The paper proposes DGS, a watermarking method for diffusion models that embeds hidden information in the latent space without retraining the model. Building on Gaussian Shading, DGS extend the method by introducing random watermark shifting and distance-weighted voting, to enhance both robustness and image diversity after watermarking. In addition, the authors  propose a new metric called Encoded Feature Diversity (EFD), for evaluating image diversity after watermarking. Empirical results demonstrate the effectiveness of the proposed method in terms of the accuracy, diversity and robustness against attack.

**Strengths:**

1. The paper focuses on an important and interesting problem in AI generative models: developing watermarking techniques to trace generated content. It indicates a key limitation of existing approaches: the loss of image diversity after watermark embedding.
2. The proposed method offers an intuitive and well-motivated solution that enhances the diversity of generated images while maintaining  robustness and watermark detection accuracy, as supported by experimental results.

**Weaknesses:**

1. A main concern of this paper is its limited novelty. The proposed method mainly builds upon the existing Gaussian Shading framework, and the key modifications including the dynamic subspace shifting and distance-weighted voting appear to be incremental extensions rather than fundamentally new concepts.
2. The baseline methods evaluated in this paper are not up-to-date. The most recent baseline included was released in April 2024. It is suggested that the authors should consider stronger and more recent baseline such as [1].

[1] Hidden in the Noise: Two-Stage Robust Watermarking for Images

3. While the proposed method demonstrates strong robustness against common attacks such as random cropping and Gaussian noise, there exist more specialized and targeted watermark removal techniques, such as [2][3][4]. It is therefore suggested that the authors evaluate their method against these advanced removal attacks to provide a more comprehensive assessment of its robustness and practical reliability.

[2] Invisible Image Watermarks Are Provably Removable Using Generative AI

[3] Generative autoencoders as watermark attackers: Analyses of vulnerabilities and threats

[4] Evading watermark based detection of AI-generated content

4. While the proposed EFD metric is intuitive and reasonable, the necessity of introducing a new diversity measure is not fully justified. It would strengthen the paper if the authors could explain why existing metrics such as Inception Score and FID are insufficient for this evaluation.
5. The paper claims that the fixed sampling subspace in Gaussian Shading causes reduced image diversity. The justification for this claim is not strong enough. Although the paper reports that Gaussian Shading yields a lower EFD score (0.18) compared to standard Stable Diffusion (0.21), the evidence provided is not sufficient to intuitively convey how significant this difference is. The paper only includes a few illustrative examples in the appendix, which do not clearly demonstrate the perceptual or semantic extent of the diversity degradation reflected by the numerical gap.
6. Small issue: The authors did not distinguish the use of \cite and \citep in in-text citations.

**Questions:**

1. In Figure 7, it seems that the people's faces are distorted in the generated watermarked images. Is it because of the existence of the watermark or simply the limited performance of the diffusion model?
2. Could the authors include ablation studies comparing the performance with and without the distance-weighted voting mechanism to demonstrate its specific contribution to the robustness and accuracy of watermark extraction?

---

### Official Review · Reviewer_KmQs · 2025-10-31

**Soundness:** 2
**Presentation:** 2
**Contribution:** 2
**Rating:** 2
**Confidence:** 3

**Summary:**

Dynamic Gaussian Shading (DGS) is a diffusion model watermarking method that introduces random subspace shifting and distance-weighted voting to achieve both high robustness and enhanced image diversity.

**Strengths:**

1. The paper presents clear and coherent logical reasoning throughout the method and analysis.

2. It successfully enhances the diversity of generated images compared to prior approaches.

3. The experimental results are comprehensive, detailed, and well-documented.

**Weaknesses:**

1. The analysis of the cause is too narrow. The paper attributes the decrease in diversity to only one factor. However, diversity degradation can result from multiple different causes. For example, in training-based watermarking, it might stem from the fine-tuning dataset being too small, leading to overfitting [1]. In latent-space-based watermarking, another important reason could be the deterministic sampling path of diffusion models—DDIM’s fixed trajectory may itself reduce randomness and thus diversity. The proposed method identifies only one of these possible causes.

2. The proposed method may simply be a special case of ID encoding utilization. For instance, given a binary string of length l, could we use l/2 of those bits purely to enhance diversity—randomly generated and interleaved with the true encoding bits? We could even let the first 10 bits specify which positions in the watermark represent the actual code versus diversity bits. This approach seems conceptually simpler than the authors’ method, since it only requires minor changes during the encoding process.

3. It is unclear why in Table 1 the post-watermarking method DwtDct shows such poor TPR (Clean) performance.

4. The adversarial evaluation is too weak. The paper does not include attacks specifically designed for watermark removal or detection evasion, such as those in [2].

5. The post-watermarking baselines are outdated. Please consider comparing with more recent and stronger methods such as [3], [4], and [5].

6. The incorrect citation formatting makes the paper almost unreadable.

7. The introduction section is overly long, especially the first two paragraphs.

8. The writing of experiment section is bad. The settings, results and analysis are combined in one subsection which makes it hard to read.

[1] A Recipe for Watermarking Diffusion Models

[2] Diffusion Models for Adversarial Purification

[3] Robust data hiding using inverse gradient attention.

[4] Mbrs: Enhancing robustness of dnn-based watermarking by mini-batch of real and simulated jpeg compression

[5] Towards blind watermarking: Combining invertible and non-invertible mechanisms.

**Questions:**

1. Experiments of the method in Weakness 2 could be provided.

2. Why in Table 1 the post-watermarking method DwtDct shows such poor TPR (Clean) performance?

3. What is the results including [2] [3], [4], and [5]?

---

### Official Review · Reviewer_F9gY · 2025-10-31

**Soundness:** 2
**Presentation:** 2
**Contribution:** 2
**Rating:** 4
**Confidence:** 3

**Summary:**

This paper proposed a text-to-image diffusion model watermarking method based on Gaussian Shading, which mostly focuses on improving diversity of the generation. This paper then proposed a metric that evaluates the diversity of the generated image based on CLIP features. After that they applied this metric for evaluating their work, along with other baseline watermarking methods, which shows that watermarking indeed reduces diversity, while their method could indeed alleviate such a problem.

**Strengths:**

1. Significance. The main contribution of this paper I think is that it addresses a concern of missing diversity after watermarking over diffusion models, and provides a reasonable metric that quantifies such an issue.
2. Structure Clarity. The paper is relatively well structured, apart from issues addressed in weakness 4 5, which downgrade the overall presentation.

**Weaknesses:**

1. Proposed method. The method does modification based on Gaussian Shading, in order to improve diversity and robustness. Although in Table 1 it indeed shows minor improvement in accuracy and robustness (adversarial), it is probably using image quality as a trade off, as indicated in FID. Perhaps additional metrics or explanations or even improvements could be done to increase competitiveness of this method.
2. Case study and visualization. Although Table 2 shows that this method has larger EFD than baselines, I think it would be better to have more case study images similar to Figure 6 and 7. Since I am interested in how each range of numeric values of EFD corresponds to the diverseness of the model. I.e. is EFD 0.21 a significant improvement than EFD 0.18 or just a minor difference? What about other values?
3. Need stronger attack. This work shows 9 attacks to evaluate the robustness of the watermark. I wonder if there are more recent, stronger attacks? How robust is this watermark against such attack(s)?
4. Ablation studies. In 5.3, experiment shows how different settings and perturbations (attacks) affect the watermark performance. It seems to be not a traditional ablation study. For clarity, is it better to use another title of this section?
5. Method Diagram. Figure 1 needs improvements of clarity. I think it is better with more hierarchy over the flowchart, with bounding boxes grouping related components together.

**Questions:**

Since model generation diversity is the highlight of this paper, we are looking for more foreshadowing about this. I wonder if diversity of diffusion model generation is already a well known issue that people believe is significant? Is it necessary to spend more words answering “why diversity is important?” in introduction?

---

### Note · Authors · 2026-01-20

I have read and agree with the venue's withdrawal policy on behalf of myself and my co-authors.